# Needs assessment and preparedness of the primary health care network for scaling-up preventive tuberculosis treatment in 5 Brazilian capitals

Dinah Carvalho Cordeiro[1,2,3], Pedro Kuabara[4], Bruna Chiarini Amaral[4], Lucas Maia Portugal[4], Daniel Souza Sacramento[2], Larissa Rodrigues de Oliveira[3], Jorgiane da Costa Ferreira Faria[3], Priscilla Wolter Paolino[3], Vanessa Cordeiro Vilanova[5], Mayara de Sá Cavalcanti de Melo[6], Cristina Bettin Waechter[7], Anete Trajman[1,3,4,8]*

1 Programa de Pós-graduação em Clínica Médica, Faculdade de Medicina, Universidade Federal do Rio de Janeiro, Rio de Janeiro, Brazil, 2 Secretaria Municipal de Saúde de Manaus, Manaus, Amazonas, Brazil, 3 Rede Brasileira de Pesquisa em Tuberculose – Rede TB, Rio de Janeiro, Brazil, 4 Graduação, Faculdade de Medicina, Universidade Federal do Rio de Janeiro, Rio de Janeiro, Brazil, 5 Secretaria Municipal de Saúde de São Paulo, São Paulo, Brazil, 6 Secretaria Municipal de Saúde de Recife, Recife, Pernambuco, Brazil, 7 Secretaria Municipal de Saúde de Porto Alegre, Porto Alegre, Rio Grande do Sul, Brazil, 8 McGill TB International Centre, McGill University, Montreal, Canadá

* atrajman@gmail.com

## Abstract

This study aims to conduct a needs and preparedness assessment of public primary care units to scale up tuberculosis infection diagnosis and tuberculosis preventive treatment in 5 Brazilian capitals. This observational operational study was carried out across five Brazilian high tuberculosis-burden cities. Clinics with at least one monthly new tuberculosis case were included. Data on Purified Protein Derivative (PPD) storage, tuberculin skin testing (TST) and interferon-gamma release assay (IGRA) availability, personnel qualified for performing TST, radiological facilities and tuberculosis preventive treatment drug availability, were gathered between August 2023 and January 2024. Out of 285 clinics included, 78% (CI95%: 73%−82%) did not offer TST on-site, with only 28% (CI95%: 22%3%) having staff qualified to perform TST, and 35% (CI95%: 29%−40%) lacking dedicated refrigerators for PPD storage. Most (97%, CI95%: 94%−99%) clinics did not collect IGRA testing, with an average distance of 6.7 km (CI95%: 5%−7%) to IGRA labs and a turnaround time of 11.7 days (CI95%: 9%13%) for results. Most (87%, CI95%: 83%−91%) do not offer on-site radiological testing. The primary care network was unprepared to scaling up tuberculosis infection testing. Key issues include unavailability of TST mainly because of insufficient qualified personnel. Without accelerated qualification of staff for TST, scaling up tuberculosis preventive treatment will be impossible.

**Data availability statement:** All relevant data are within the paper and its Supporting Information files.

**Funding:** Stop TB Partnership, TB REACH wave 10, project #10429. The funders had no role in study design, data collection and analysis, decision to publish, or preparation of the manuscript.

**Competing interests:** The authors have declared that no competing interests exist.

## Introduction

Tuberculosis preventive treatment is a cornerstone of the pathway to tuberculosis elimination [1]. The 2023 United Nations High-Level Meeting on the fight against tuberculosis committed to providing 45 million courses of tuberculosis preventive treatment until 2027 [2]. This is an ambitious target, as the world has not yet achieved the goal set by the United Nations High-Level Meeting, held in 2018 [3], to provide 30 million courses of tuberculosis preventive treatment. The least progress was among contacts of people with pulmonary tuberculosis, especially those over 5 years of age [4].

Some advancements have been made in tests for tuberculosis infection in the past decade, and shorter and safer tuberculosis preventive treatment regimens are now available [4]. However, several obstacles remain to achieving the United Nations High-Level Meeting goals. These obstacles include the loss of tuberculosis infection contacts at various steps of the cascade of care [5], often due to factors such as client and healthcare worker beliefs [6–8], need for testing, and stockouts of Purified Protein Derivative [9], among others.

The Brazilian National Tuberculosis Program has recommended, among other indications, treating tuberculosis contacts of all ages since 2010. With a few specific exceptions, contacts undergo a tuberculin skin testing (TST) and if positive (≥5mm), they undergo chest radiography (CXR). interferon-gamma release assays (IGRA) are available for specific populations, such as children (2–10 years old), people living with HIV or using immunosuppressive therapy and transplant candidates. If there are no radiological signs of tuberculosis, tuberculosis preventive treatment is prescribed. Currently, the first-choice regimen is 3HP (12 weekly doses of rifapentine and isoniazid) for those between 10 and 50 years of age, and 3HR (3 months of daily isoniazid and rifampicin) for those under 10. For those over 50 years of age or with underlying liver disease, 4R (4 months of rifampicin) is the recommended regimen. All tests and treatments are free of charge [10,11].

In 2023, Brazil reported 80,012 new cases of tuberculosis, of which 70,273 were pulmonary. According to the country's demographics, there are 2.7 inhabitants per household on average. This translates to an average of 1.7 household (resident) contacts. We would expect at least 119,464 contacts initiating evaluation for tuberculosis and tuberculosis infection, and with an expected 50%−60% eligibility for tuberculosis preventive treatment, at least 60,000 individuals should start tuberculosis preventive treatment yearly. Instead, in 2023, 20,365 contacts initiated tuberculosis preventive treatment in the country [10–12]. The main losses in the cascade of care still occur during testing for tuberculosis infection: over 50% of the identified contacts do not complete initial evaluation with a tuberculosis infection test [13]. Initial training in the ExpandTPT program resulted in a substantial increase in the identification of contacts, but not in the proportion of contacts tested. This could be due to contacts not presenting for testing or healthcare workers not offering the necessary tests.

Thus, despite the recommendations, tuberculosis preventive treatment prescription remains slow in the country. In close partnership with the Brazilian National Tuberculosis Program, the Brazilian Tuberculosis Research Network (www.redetb.org.br), in

collaboration with the McGill International Tuberculosis Centre, conducted the ExpandTPT project [14] to scale up tuberculosis preventive treatment in five high tuberculosis-burden cities. We conducted a baseline needs assessment to scale up tuberculosis infection diagnosis and tuberculosis preventive treatment of public primary care units with the highest monthly numbers of new tuberculosis diagnosed in these cities.

## Materials and methods

### Study design

This is a descriptive observational operational study.

### Setting and inclusion criteria

Five cities were selected to participate in the ExpandTPT program by the Brazilian National Tuberculosis Program: Manaus, Recife, Rio de Janeiro, Porto Alegre and São Paulo. Their incidence rates of tuberculosis in 2023 were 113.2; 99.0; 93.4; 64.8 and 58.3/100,000 population, respectively. While their total number of pulmonary tuberculosis reported in 2023 were 2,312, 1,418, 5,698, 814, e 6,035, the total number of tuberculosis preventive treatment prescribed in the same year were 1,551, 609, 3,158, 698 and 5,892, respectively [12].

Evaluation of contacts and tuberculosis preventive treatment are conducted in primary care units in Brazil. There are 1,193 clinics in the 5 cities selected, 845 are Family Health Clinics and 348 are "traditional" clinics. In Family Health Clinics, each family health team is composed by one medical doctor, one registered nurse and 6–10 community health workers and is responsible for the health of 1000 families, or, on average, 4000 persons, offering prevention activities and health promotion, with home visits besides the services at the facility. In traditional clinics, clients spontaneously seek for care, eventually there are tuberculosis/pulmonology services, or it can be offered by a generalist medical doctor [15,16].

Tuberculin skin testing is offered in primary care units, and IGRA in labs, although blood may be withdrawn in the clinic and forwarded to the reference lab by the health facility. Purified Protein Derivative (PPD) is procured by the Ministry of Health, then distributed to states, that distribute to cities, and municipal clinics store in fridges dedicated to medicines. The Ministry of Health Immunization Program does not permit storing PPD with the vaccines [17]. Three months of weekly doses of rifapentine and isoniazid may be offered under direct observation or self-administered. Drugs are acquired by the Ministry of Health and distributed to states and cities, public health clinics have a pharmacy to store and distribute treatment free of charge.

Clinics located in urban areas that serve, on average, at least one person with newly diagnosed tuberculosis per month were selected to be included in the ExpandTPT project, and included in this survey.

### Data gathering and analysis

Data on PPD storage, TST availability at the clinic, qualified personnel for (TST) injection and reading, weekdays when TST is offered, IGRA availability, radiological facilities, 3HP (3 months of weekly doses of rifapentine and isoniazid) availability, notification and work process were gathered through questionnaires applied to the clinic directors S1 File. Data were collected between August 2023 and January 2024 by the study team. Data were obtained from clinics personnel. Informants were waived from informed consent by the ethical committee. A descriptive analysis was conducted, stratified by city. Percentages were calculated, excluding missing data from the denominator. The database urderlying the findings of this study is provided in S2 File.

### Ethics

Municipal tuberculosis managers authorized and supervised the data collection. The project was approved by the Ethical Committees of *Hospital Universitário Clementino Fraga Filho* (**5.936.814**, Mar 10, 2023), *Secretaria Municipal de Saúde*

*do Rio de Janeiro* (**5.977.938** – Mar 31, 2023), *Secretaria Municipal de Saúde de São Paulo* (**6.019.984** – Apr 24, 2023), Secretaria *Municipal de Saúde de Porto Alegre* (**6.434. 607** – Oct 19, 2023), *Secretaria Municipal de Saúde de Recife* (**n°, 0425/2023** – DES/GFES/SEGTES/SESAU, Mar 17, 2023) and authorized by *Secretaria Municipal de Saúde de Manaus* (**n°. 08** – ESAP/SEMSA, Mar 16, 2023).

## Results and discussion

Two hundred and eighty-five clinics fulfilled the criteria and were included in the study, of which 168 were Family Health Clinics and 117 were traditional clinics.

Table 1 summarizes the needs assessment findings. Most clinics 221/283 (78%, 95%CI: 73%−82%) did not offer TST on-site; they referred contacts to other clinics. Of the 62 clinic that offered TST, only three offered it daily. Most 52/62 (84%, 95%CI: 74%−93%) offered the test from once to three times weekly. Only one of the 62 clinics offered TST on demand, i.e., without need to schedule. Other findings related to TST were the low number of health care workers qualified to read TST induration in the unit 80/285 (28%, 95%CI: 22%−33%), and the large number of units without a dedicated fridge to store PPD 98/28 (35%, 95%CI: 29%−40%). Tuberculosis blood test IGRA is generally not collected onsite 151/156 (97%, 95%CI: 94%−99%), contacts are referenced to a laboratory where the test is available. The mean distance

**Table 1. Needs assessment for scaling up tuberculosis preventive treatment in 285 primary care clinics in 5 high burden cities in Brazil (n = number of clinics assessed)[a].**

| Activity/infrastructure | São Paulo (n=58) | Rio de Janeiro (n=47) | Manaus (n=53) | Recife (n=45) | Porto Alegre (n=82) | Total (n=285) n (%) | 95%CI |
|---|---|---|---|---|---|---|---|
| Tuberculin skin testing offered at the Clinic | | | | | | | |
| Yes | 7 (12%) | 30 (64%) | 16 (30%) | 4 (9%) | 5 (6%) | 62 (22%) | 17–26% |
| No | 51 (88%) | 17 (36%) | 37 (70%) | 39 (91%) | 77(94%) | 221 (78%) | 73–82% |
| Health care worker at the clinic qualified for tuberculin skin testing reading | | | | | | | |
| Yes | 12 (21%) | 33 (70%) | 19 (36%) | 6 (13%) | 10 (12%) | 80 (28%) | 22–33% |
| No | 46 (79%) | 14 (30%) | 34 (64%) | 39 (87%) | 72 (88%) | 205 (72%) | 66–77% |
| Fridge for Purified Protein Derivative storage | | | | | | | |
| Yes | 58 (100%) | 42 (93%) | 43 (81%) | 11 (26%) | 29 (35%) | 183 (65%) | 59–70% |
| No | 0 (0%) | 3 (7%) | 10 (19%) | 32 (74%) | 53 (65%) | 98 (35%) | 29–40% |
| Blood withdrawal for interferon-gamma release assay (n=156)[b] | | | | | | | |
| Yes | 1 (2%) | - | 4 (8%) | 0 (0%) | - | 5 (3%) | 0-6% |
| No | 57 (98%) | | 49 (92%) | 45 (100%) | | 151 (97%) | 94–99% |
| Mean distance for interferon-gamma release assay lab)[c] | 7.7km n=50 | - | 5.6km n=49 | - | - | 6.7km (1.4km - 45km - n=99) | 5–7% |
| Mean time for interferon-gamma release assay results[c] | 11.2 days n=22 | - | 10.2 days n=39 | - | - | 11,7 days (2 - 45 days - n=61) | 9–13% |
| Chest X-Ray equipment at the clinic | | | | | | | |
| Yes | 4 (7%) | 25 (53%) | 6 (11%) | 1 (2%) | 0 (0%) | 36 (13%) | 8–16% |
| No | 54 (93%) | 22 (47%) | 47 (90%) | 44 (98%) | 82 (100%) | 249 (87%) | 83–91% |
| 3 months of weekly doses of rifapentine and isoniazid is the first-choice regimen for adults | | | | | | | |
| Yes | 41 (71%) | 39 (89%) | 48 (91%) | 18 (40%) | 61 (74%) | 207 (73%) | 68–78% |
| No | 17 (29%) | 5 (11%) | 5 (9%) | 27 (60%) | 21 (26%) | 75 (27%) | 21–31% |

[a]Percentages calculated over valid numbers

[b]Rio de Janeiro and Porto Alegre did not collect this information

[c]Rio de Janeiro, Recife and Porto Alegre did not collect this information

between the referenced laboratory and clinic is 6.7 km (95%CI: 5%−7%), the mean turnaround time for IGRA results is 11.7 days (95%CI: 9%−13%).

With regards to tuberculosis exclusion, most clinics 249/285 (87%, 95%CI: 83%91%) do not have radiologic equipment onsite. Clients are referred to other primary or secondary clinics or use private radiologic services (paid by the government).

Our findings in the current study suggest that at least part of this initial loss in household contact assessment can be attributed to health services limitations. Clinics do not offer TST, a simple and cost-effective test that could be available in all primary care clinics [18] or perhaps in an area where TB was very decentralized or had fewer cases, at least be offered closer to home. Another potential solution for units with small number of TB diagnosis would be the procurement of PPD vials with one or at most five doses. The scarcity of personnel considered qualified to perform TST is partially a consequence of the stringent criteria that has been used in the last decades to certify healthcare workers in Brazil: they needed to read at least 100 indurations with an 80% concordance with the trainer [19]. However, based on data from the ExpandTPT project and other evidence, these training recommendations have recently been updated, with the aim of expanding the qualification of professionals throughout the country [20].

There was heterogeneity in the number of clinics offering TST across the cities. In Rio de Janeiro, we found a greater number of professionals and clinics offering the test. This scenario may be explained by the conduct of multiple research projects on latent tuberculosis infection in the city with the intensification of local training, but the test still needs to be expanded.

In addition, the need for a dedicated fridge is a bottleneck in some small clinics in the rural areas. The National Immunization Program recommendation that prohibits the storage of PPD with vaccines is unjustified [19]. Finally, recurrent stockouts of PPD jeopardize training and implementation of more widespread testing in the country [9,21].

Based on these findings, the ExpandTPT project, in partnership with the Brazilian MoH, is offering training on the National Guidelines and increasing the TST network to a wide range of clinics by training more healthcare workers. In our initial assessment of the cascade of care in these cities, exclusion of tuberculosis with chest X-rays was not a substantial barrier: of those with a positive tuberculosis infection test, 80% underwent a chest X-ray, but the expansion of the TST network may have an impact on the availability of chest X-rays for all eligible contacts.

The World Health Organization has recommended offering tuberculosis preventive treatment without testing in countries where tests are not available. However, a recent individual patient metanalysis has shown that tuberculosis preventive treatment is beneficial for those who have a positive test, both in high-income low-tuberculosis burden and low-middle income high-burden countries [22]. Brazil has a structured public health system, and national recommendations guide testing, through the tuberculin skin test or IGRA, to direct preventive tuberculosis treatment to those who present positive results and are at greater risk of illness [10,11].

Our study has limitations. Self-reported data by clinic staff may potentially generate bias. Thus, true data may be still worse. Another limitation is the absence of health system users' perspective that could influence adherence to testing and tuberculosis preventive treatment uptake.

## Conclusions

In conclusion, the primary care network was not ready, at the time of this evaluation, to scale up testing for tuberculosis infection and thus offering tuberculosis preventive treatment. Softening of training, availability of PPD or the newer tuberculosis-specific skin tests approved by the World Health Organization, and a widespread cold chain are necessary to scale up tuberculosis preventive treatment in the country.

## Supporting information

**S1 File. Survey instrument used for data collection.**
(PDF)

**S2 File. Raw data underlying the study findings.**
(XLSX)

## Author contributions

**Conceptualization:** Priscilla Wolter Paolino, Anete Trajman.

**Data curation:** Dinah Carvalho Carvalho Cordeiro, Pedro Kuabara, Bruna Chiarini Amaral, Lucas Maia Portugal, Daniel Souza Sacramento, Larissa Rodrigues de Oliveira, Jorgiane da Costa Ferreira Faria, Priscilla Wolter Paolino, Vanessa Cordeiro Vilanova, Mayara de Sá Cavalcanti de Melo, Cristina Bettin Waechter, Anete Trajman.

**Formal analysis:** Dinah Carvalho Carvalho Cordeiro, Pedro Kuabara, Bruna Chiarini Amaral, Lucas Maia Portugal, Daniel Souza Sacramento, Priscilla Wolter Paolino, Vanessa Cordeiro Vilanova, Mayara de Sá Cavalcanti de Melo, Cristina Bettin Waechter, Anete Trajman.

**Funding acquisition:** Anete Trajman.

**Investigation:** Daniel Souza Sacramento, Larissa Rodrigues de Oliveira, Jorgiane da Costa Ferreira Faria, Priscilla Wolter Paolino, Vanessa Cordeiro Vilanova, Mayara de Sá Cavalcanti de Melo, Cristina Bettin Waechter, Anete Trajman.

**Methodology:** Daniel Souza Sacramento, Priscilla Wolter Paolino, Anete Trajman.

**Project administration:** Priscilla Wolter Paolino, Anete Trajman.

**Resources:** Anete Trajman.

**Supervision:** Dinah Carvalho Carvalho Cordeiro, Daniel Souza Sacramento, Larissa Rodrigues de Oliveira, Jorgiane da Costa Ferreira Faria, Priscilla Wolter Paolino, Vanessa Cordeiro Vilanova, Mayara de Sá Cavalcanti de Melo, Cristina Bettin Waechter, Anete Trajman.

**Visualization:** Priscilla Wolter Paolino, Anete Trajman.

**Writing – original draft:** Dinah Carvalho Carvalho Cordeiro, Pedro Kuabara, Bruna Chiarini Amaral, Lucas Maia Portugal, Daniel Souza Sacramento, Larissa Rodrigues de Oliveira, Jorgiane da Costa Ferreira Faria, Priscilla Wolter Paolino, Vanessa Cordeiro Vilanova, Mayara de Sá Cavalcanti de Melo, Cristina Bettin Waechter, Anete Trajman.

**Writing – review & editing:** Dinah Carvalho Carvalho Cordeiro, Pedro Kuabara, Bruna Chiarini Amaral, Lucas Maia Portugal, Daniel Souza Sacramento, Larissa Rodrigues de Oliveira, Jorgiane da Costa Ferreira Faria, Priscilla Wolter Paolino, Vanessa Cordeiro Vilanova, Mayara de Sá Cavalcanti de Melo, Cristina Bettin Waechter, Anete Trajman.

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
