## [Decision Letter · Decision Letter 0]

PONE-D-25-04795Needs assessment and preparedness of the primary health care network for scaling-up preventive tuberculosis treatment in 5 Brazilian capitalsPLOS ONE

Dear Dr. Cordeiro,

Thank you for submitting your manuscript to PLOS ONE. After careful consideration, we feel that it has merit but does not fully meet PLOS ONE’s publication criteria as it currently stands. Therefore, we invite you to submit a revised version of the manuscript that addresses the points raised during the review process.

Please submit your revised manuscript by May 01 2025 11:59PM. If you will need significantly more time to complete your revisions, please reply to this message or contact the journal office at plosone@plos.org . Please include the following items when submitting your revised manuscript:

We look forward to receiving your revised manuscript.

Kind regards,

Frederick Quinn

Academic Editor

PLOS ONE

Journal Requirements:

Stop TB Partnership, TB REACH wave 10, project #10429.

4. We note that your Data Availability Statement is currently as follows: Todos os dados relevantes estão dentro do manuscrito e de seus arquivos de Informações de Suporte.

Reviewers' comments:

Reviewer's Responses to Questions

**Comments to the Author**

1. Is the manuscript technically sound, and do the data support the conclusions?

Reviewer #1: Yes

Reviewer #2: Partly

2. Has the statistical analysis been performed appropriately and rigorously? 

Reviewer #1: Yes

Reviewer #2: No

3. Have the authors made all data underlying the findings in their manuscript fully available?

Reviewer #1: No

Reviewer #2: Yes

4. Is the manuscript presented in an intelligible fashion and written in standard English?

Reviewer #1: Yes

Reviewer #2: Yes

5. Review Comments to the Author

Reviewer #1: Very interesting paper over implementation of interventions for a clear PH problem. Please see my suggestions/ remarks for consideration

Settings

1. Could it be clarified if the different regions researched are different in looking at urban, rural facilities

Results

2. Please consider removing everywhere the ‘excluding missing data”. With the nominator and denominator and mention in the methods is already clear that missing data is excluded and now it is sometimes mentioned where there even is no data missing.

3. There are difference seen in the results of the different areas. Can there be more mention of that in the results. Are there differences in rural versus urban?

Discussion

4. I suggest the first paragraph of the discussion be moved to the introduction. It more clearly explains the need to implement these strategies and why this assessment than is so important

5. There is mention of little personnel skilled to perform TST due to the stringent criteria. I miss the discussion whether this is fair or how this could be dealt with

6. Consider discussing the different aspects looked at separately.

7. Please clarify the point regarding stockouts of PPD jeopardizing training

8. The point regarding WHO recommendations to offer TB preventive treatment without tests and the following paragraph is not clear. Could authors please clarify what they are recommending for their country

9. Consider elaborating in the discussion about the different findings by region. With some clear recommendations

Reviewer #2: Title: Needs assessment and preparedness of the primary health care network for scaling-up preventive tuberculosis treatment in 5 Brazilian capitals

The manuscript is well-written and effectively addresses the challenges of scaling up tuberculosis preventive treatment (TPT) in five high-burden Brazilian cities through a descriptive and observational study design, suitable for identifying implementation gaps.

Methodology: The variation in TPT prescription rates suggests disparities in implementation across cities, possibly due to differences in healthcare infrastructure, resource allocation, or adherence to national guidelines. I wonder if the authors collected other factors that could drive these disparities and develop targeted interventions to ensure equitable TPT access across all high-burden areas.

The inclusion of the questionnaire as supplemental material would be valid and enable others to understand the specific questions asked, the structure of the survey, and the context in which responses were gathered.

Statistical analysis is missing, only descriptive analysis was conducted, stratified by city, with percentages calculated excluding missing data.

Discussion: as a suggestion, some points that can be included in the discussion to make it more robust include 1. consider the nature of the study, limiting the ability to assess trends or long-term intervention 2. the potentially biased self-reported data from clinic directors and the absence of patient perspectives, such as stigma or awareness issues, which could influence TPT uptake.

Table 1: please correct alignment of the row 3 (Health…), 5 (Blood…) and 9 (3 months…)

6. PLOS authors have the option to publish the peer review history of their article (what does this mean? ). If published, this will include your full peer review and any attached files.

**Do you want your identity to be public for this peer review?** For information about this choice, including consent withdrawal, please see our Privacy Policy .

Reviewer #1: No

Reviewer #2: No

---

## [Author Response · Author response to Decision Letter 1]

6 May 2025

PONE-D-25-04795

Needs assessment and preparedness of the primary health care network for scaling-up preventive tuberculosis treatment in 5 Brazilian capitals.

Dear Dr. Frederick Quinn

Editor, PLOS ONE

We are pleased to submit for your consideration a revised version of our manuscript entitled Needs assessment and preparedness of the primary health care network for scaling-up preventive tuberculosis treatment in 5 Brazilian capitals. You will find below a point-by-point reply to the reviewers’ comments. We believe that their suggestions were very useful to improve our manuscript and hope this revised version is suitable for publication in Plos One.

All authors have approved this revised version of the manuscript and take full responsibility for all its aspects.

We certify that this manuscript is original, has not been published in whole or in part, and is not under consideration for publication in any other journal, whether in print or electronic format.

We declare that there are no conflicts of interest related to this work.

Dinah Carvalho Cordeiro, on behalf of all authors

Journal Requirements:

Response: Thank you for reviewing our manuscript and for your suggestions to improve it further.

Response: We have revised the formatting of the manuscript to meet PLOS ONE’s requirements.

Response: We revised.

Stop TB Partnership, TB REACH wave 10, project #10429.

Response: We included the text “The funders had no role in study design, data collection and analysis, decision to publish, or preparation of the manuscript” in the cover letter.

4. We note that your Data Availability Statement is currently as follows: Todos os dados relevantes estão dentro do manuscrito e de seus arquivos de Informações de Suporte.

Response: We are submitting the database as supplementary information and have requested Rede-TB to host the database as well .

Response: All references have been reviewed to ensure they follow the Vancouver style, and access links have been verified.

Reviewers' comments:

Reviewer's Responses to Questions

Comments to the Author

1. Is the manuscript technically sound, and do the data support the conclusions?

Reviewer #1: Yes

Reviewer #2: Partly

2. Has the statistical analysis been performed appropriately and rigorously?

Reviewer #1: Yes

Reviewer #2: No

3. Have the authors made all data underlying the findings in their manuscript fully available?

Reviewer #1: No

Reviewer #2: Yes

4. Is the manuscript presented in an intelligible fashion and written in standard English?

Reviewer #1: Yes

Reviewer #2: Yes

5. Review Comments to the Author

Reviewer #1: Very interesting paper over implementation of interventions for a clear PH problem. Please see my suggestions/ remarks for consideration

Response: Thank you for your encouragemen and for your very useful suggestionst.

Settings

1. Could it be clarified if the different regions researched are different in looking at urban, rural facilities

Response: We clarified that the clinics were located in urban áreas (lines 94-96).

Results

2. Please consider removing everywhere the ‘excluding missing data”. With the nominator and denominator and mention in the methods is already clear that missing data is excluded and now it is sometimes mentioned where there even is no data missing.

Response: We have excluded.

3. There are difference seen in the results of the different areas. Can there be more mention of that in the results. Are there differences in rural versus urban?

Response: We were not able to evaluate if there are differences between rural and urban areas, since the selected clinics were all in urban areas, but we did observe differences between cities. Rio de Janeiro had more professionals qualified and more clinics that offered the tuberculin skin test. This may be due to the fact that some research on latent tuberculosis was conducted in Rio de Janeiro and, at that time, there was intensive training in performing the tuberculin skin test.We clarify this in the results and discussion. We have added that to the text (lines 148-151).

Discussion

4. I suggest the first paragraph of the discussion be moved to the introduction. It more clearly explains the need to implement these strategies and why this assessment than is so importante

Response: We have moved as suggested.

5. There is mention of little personnel skilled to perform TST due to the stringent criteria. I miss the discussion whether this is fair or how this could be dealt with

Response: We included this information in the manuscript (lines 145-147). Recent changes to national recommendations for training health professionals in tuberculin skin test were published by the Ministry of Health in December 2024. These changes were based on data from the ExpandTPT project and other evidence. The change is now one of the references.

6. Consider discussing the different aspects looked at separately.

Response: We have summarized the findings in the first paragraph and have discussed diferente aspects separately.

7. Please clarify the point regarding stockouts of PPD jeopardizing training

Response: We have included in the text (lines154-155). We could not use the remaining doses of PPD for training as we saved it for the situations in which it eas most needed. We hope this is now clear.

8. The point regarding WHO recommendations to offer TB preventive treatment without tests and the following paragraph is not clear. Could authors please clarify what they are recommending for their country

Response: We have rephrased (lines 166-168).

9. Consider elaborating in the discussion about the different findings by region. With some clear recommendations

Response: Our recommendations are general, because although Rio de Janeiro has more clinics offering the tuberculin skin test, it still needs to expand the offer of the test. We have includde this in the discussion (lines 148-151).

Reviewer #2: Title: Needs assessment and preparedness of the primary health care network for scaling-up preventive tuberculosis treatment in 5 Brazilian capitals

The manuscript is well-written and effectively addresses the challenges of scaling up tuberculosis preventive treatment (TPT) in five high-burden Brazilian cities through a descriptive and observational study design, suitable for identifying implementation gaps.

Response: Thank you for your review and for your suggestions. We believe they have improved the manuscript.

Methodology: The variation in TPT prescription rates suggests disparities in implementation across cities, possibly due to differences in healthcare infrastructure, resource allocation, or adherence to national guidelines. I wonder if the authors collected other factors that could drive these disparities and develop targeted interventions to ensure equitable TPT access across all high-burden areas.

Response: The main difference between the cities is the availability of tuberculin skin testing. We believe this is due to the intensive training for preventive treatment research in Rio de Janeiro. We have added this discussion in lines 148-151.

The inclusion of the questionnaire as supplemental material would be valid and enable others to understand the specific questions asked, the structure of the survey, and the context in which responses were gathered.

Response: We have included our questionnaire as supplementary material.

Statistical analysis is missing, only descriptive analysis was conducted, stratified by city, with percentages calculated excluding missing data.

Response: We did not intend an analytical approach to compare the five cities because the numbers were small; the descriptive analysis was sufficient for our conclusions. We have now included the 95%CI in the text and table.

Discussion: as a suggestion, some points that can be included in the discussion to make it more robust include 1. consider the nature of the study, limiting the ability to assess trends or long-term intervention 2. the potentially biased self-reported data from clinic directors and the absence of patient perspectives, such as stigma or awareness issues, which could influence TPT uptake.

Response: Thank you for these suggestions. We have included these points as limitations: “Our study has limitations. Self-reported data by clinic staff may potentially generate bias. Thus, true data may be still worse. Another limitation is the absence of health system users’ perspective that could influence adherence to testing and tuberculosis preventive treatment uptake”.

Table 1: please correct alignment of the row 3 (Health…), 5 (Blood…) and 9 (3 months…)

Response: We have corrected.

6. PLOS authors have the option to publish the peer review history of their article (what does this mean?). If published, this will include your full peer review and any attached files.

Do you want your identity to be public for this peer review? For information about this choice, including consent withdrawal, please see our Privacy Policy.

Reviewer #1: No

Reviewer #2: No

---

## [Decision Letter · Decision Letter 1]

Needs assessment and preparedness of the primary health care network for scaling-up preventive tuberculosis treatment in 5 Brazilian capitals

PONE-D-25-04795R1

Dear Dr. Cordeiro,

We’re pleased to inform you that your manuscript has been judged scientifically suitable for publication and will be formally accepted for publication once it meets all outstanding technical requirements.

Kind regards,

Frederick Quinn

Academic Editor

PLOS ONE

Additional Editor Comments (optional):

Reviewers' comments:

Reviewer's Responses to Questions

**Comments to the Author**

1. If the authors have adequately addressed your comments raised in a previous round of review and you feel that this manuscript is now acceptable for publication, you may indicate that here to bypass the “Comments to the Author” section, enter your conflict of interest statement in the “Confidential to Editor” section, and submit your "Accept" recommendation.

Reviewer #1: All comments have been addressed

2. Is the manuscript technically sound, and do the data support the conclusions?

Reviewer #1: Yes

3. Has the statistical analysis been performed appropriately and rigorously? 

Reviewer #1: Yes

4. Have the authors made all data underlying the findings in their manuscript fully available?

Reviewer #1: Yes

5. Is the manuscript presented in an intelligible fashion and written in standard English?

Reviewer #1: Yes

6. Review Comments to the Author

Reviewer #1: (No Response)

7. PLOS authors have the option to publish the peer review history of their article (what does this mean? ). If published, this will include your full peer review and any attached files.

**Do you want your identity to be public for this peer review?** For information about this choice, including consent withdrawal, please see our Privacy Policy .

Reviewer #1: No

---

## [Editor Report · Acceptance letter]

PONE-D-25-04795R1

PLOS ONE

Dear Dr. Cordeiro,

I'm pleased to inform you that your manuscript has been deemed suitable for publication in PLOS ONE. Congratulations! Your manuscript is now being handed over to our production team.

Kind regards,

on behalf of

Dr. Frederick Quinn

Academic Editor

PLOS ONE